# The Ubiquitin Ligase RNF138 Cooperates with CtIP to Stimulate Resection of Complex DNA Double-Strand Breaks in Human G1-Phase Cells

**DOI:** 10.3390/cells11162561

**Published:** 2022-08-17

**Authors:** Nicole B. Averbeck, Carina Barent, Burkhard Jakob, Tatyana Syzonenko, Marco Durante, Gisela Taucher-Scholz

**Affiliations:** 1Department of Biophysics, GSI Helmholtzzentrum für Schwerionenforschung GmbH, Planckstr. 1, 64291 Darmstadt, Germany; 2Department of Biology, Technische Universität Darmstadt, Schnittspahnstr. 11, 64287 Darmstadt, Germany; 3Department of Condensed Matter Physics, Technische Universität Darmstadt, Hochschulstr. 6–8, 64289 Darmstadt, Germany

**Keywords:** DNA double-strand break (DSB), complex DSBs, DSB resection, ubiquitination, RNF138, CtIP, heavy ions, radiotherapy

## Abstract

DNA double-strand breaks (DSBs) represent the molecular origin of ionizing-radiation inflicted biological effects. An increase in the ionization density causes more complex, clustered DSBs that can be processed by resection also in G1 phase, where repair of resected DSBs is considered erroneous and may contribute to the increased biological effectiveness of heavy ions in radiotherapy. To investigate the resection regulation of complex DSBs, we exposed G1 cells depleted for different candidate factors to heavy ions or α-particle radiation. Immunofluorescence microscopy was used to monitor the resection marker RPA, the DSB marker γH2AX and the cell-cycle markers CENP-F and geminin. The Fucci system allowed to select G1 cells, cell survival was measured by clonogenic assay. We show that in G1 phase the ubiquitin ligase RNF138 functions in resection regulation. RNF138 ubiquitinates the resection factor CtIP in a radiation-dependent manner to allow its DSB recruitment in G1 cells. At complex DSBs, RNF138′s participation becomes more relevant, consistent with the observation that also resection is more frequent at these DSBs. Furthermore, deficiency of RNF138 affects both DSB repair and cell survival upon induction of complex DSBs. We conclude that RNF138 is a regulator of resection that is influenced by DSB complexity and can affect the quality of DSB repair in G1 cells.

## 1. Introduction

The biological effects of ionizing radiation strongly depend on radiation-induced DNA damage, in particular DNA double-strand breaks (DSBs). Erroneous or missing repair of these lesions can lead to carcinogenesis and cell death. For repairing DSBs, cells have evolved several repair pathways of different accuracy [1,2]. The two major mechanisms are Homologous Recombination (HR) and Non-Homologous End Joining (NHEJ), specifically the so called classical NHEJ (c-NHEJ). HR uses sister chromatids as a template for accurate repair and occurs only in S and G2 phase. C-NHEJ on the other hand operates throughout the cell cycle and repairs the majority of DSBs [3]. This pathway is strongly affected by the break’s structure and cannot readily deal with complex DNA damage [4,5,6].

Ionizing radiation and radiomimetic compounds can cause complex DNA damage, i.e., damage clusters, which comprise of two or more lesions within one to two helical turns of DNA [7,8]. With increasing Linear Energy Transfer (LET) and therefore ionization density of radiation, the density of DNA damage and thus damage complexity increases [9,10]. X- and γ-rays represent low-LET radiation while ions like α-particles, heavy ions or low energetic protons are high-LET radiation [11]. A special feature of complex DNA damage induced by high-LET radiation are clusters of DSBs, which are difficult to repair [6,11,12,13,14,15,16,17] and are far more frequent upon high-LET than low-LET irradiation [11]. Based on the damage structure, DSBs within complex DNA damage frequently require break-end processing in order to become repaired [18]. An important mechanism of end-processing is resection since it impacts on the DSB repair-pathway choice. With increasing damage complexity, DSBs become increasingly subject to resection [19,20] and thus are no longer a direct substrate of c-NHEJ. Interestingly, this occurs not only in G2-phase cells, where resection-dependent HR occurs, but also in G1-phase cells [19,20] in which HR does not operate and resection-dependent repair pathways are expected to be error-prone [21]. As a result, resection may be a source of genomic instability upon induction of complex DNA damage. Consequently, it may represent the basis for the increased biological effectiveness of heavy-ion radiation that is exploited in ion-beam tumor therapy [21,22]. To further clarify the mechanisms underlying the molecular response to complex DNA damage, we study the regulation of resection in G1-phase cells upon irradiation with α-particles or heavy ions.

One important layer of resection regulation in the context of the DNA-damage response is performed by 53BP1 and its binding factor RIF1, which block resection in G1 cells. In G2-phase cells, BRCA1 together with the resection factor CtIP is required to overcome this block [23,24,25]. In addition, ubiquitination has emerged as an important post translational modification (PTM) that coordinates DSB repair [26] and within this context also resection [27]. This modification involves the attachment of either single (monoubiquitination) or multiple (polyubiquitination) ubiquitin moieties to lysine residues of the target protein. Ubiquitination occurs in three main steps that each requires specific enzymes; ubiquitin activation (E1 ubiquitin-activating enzymes), ubiquitin conjugation to a ubiquitin ligase (E2 ubiquitin-conjugating enzymes), and lastly ubiquitin ligation where ubiquitin or a ubiquitin chain is attached to a target protein (E3 ubiquitin ligases) [28].

Ubiquitination within DNA repair and resection regulation controls on the one hand the recruitment of specific repair- and resection-relevant factors, e.g., 53BP1 [29,30], by modifying histones close to DSBs [31]. On the other hand, resection relevant factors become ubiquitinated themselves in order to orchestrate their presence at DSBs, e.g., Ku80, which can impede resection, and the resection factor CtIP [27]. RNF8 and RNF138 are E3 ubiquitin ligases that participate in resection regulation. RNF138 is involved in resection prior to HR in G2 phase; its ubiquitination function is needed to remove the resection antagonist Ku80 from DSBs and to recruit the resection factor CtIP to DSBs [32,33]. RNF8 is necessary for removing Ku80 from DSB sites in the course of NHEJ in G1-phase cells [32,34] and further to promote resection prior to HR in G2 phase by removing RYBP from DSB sites [35]. In addition, recruitment of RNF8 to complex DSBs was observed after α-particle irradiation [36].

In our study, we focused on resection-relevant processes at complex DSBs in G1-phase cells. Our data suggest that 53BP1/RIF1 is no major obstacle to prevent resection at these lesions. We found that in G1-phase cells the E3 ubiquitin ligase RNF138 is essential for the recruitment of the resection factor CtIP to complex heavy-ion-induced DSBs and thus important for resection regulation at these damage sites.

## 2. Materials and Methods

### 2.1. Cell Lines, Cell Culture, Transfection, and Clonogenic Survival Assay

The human osteosarcoma cell line U2OS (ATCC, Manassas, VA, USA), the human immortalized fibroblast cell line 82-6hTERT [37], the human immortalized Artemis deficient cell line CJ179hTERT [38], and the human cervix carcinoma cell line HeLa.S-Fucci (RCB2812) [39] were grown in Dulbecco’s Modified Eagle Medium (DMEM) supplemented with 10% fetal calf serum (FCS). U2OS-RNF138-KO and HeLa.S-Fucci-RNF138-KO were kept in DMEM, 10% FCS, 1 µg/mL Puromycin. The HeLa.S-Fucci cell lines are equipped with Fucci (fluorescent ubiquitination-based cell cycle indicator) that allows identifying a cell’s status of the cell cycle. The HeLa.S-Fucci cell line (RCB2812) was provided by the RIKEN BRC through the National Bio-Resource Project of the MEXT, Japan. Based on a material transfer agreement, these cells and their derivative HeLa.S-Fucci-RNF138-KO cannot be distributed by the authors. NFFhTERT cells, immortalized human fibroblasts, were kept in DMEM with 15% FCS. All cells were incubated in a humidified incubator at 37 °C and 5% CO_2_. The U2OS and HeLa.S-Fucci RNF138 knockout cell lines were obtained following the protocol of Ran*,* et al. [40]; the RNF138 specific sequence within the sgRNA was ATCATCTTCGGTGTAGGACG. Gene knockdowns were performed with HiPerfect (Qiagen, Hilden, Germany) according to the manufacturer’s protocol; the incubation time was 48 h. For the RNF138 knockdown we used RNF138 specific siRNA1 [32] and siRN138-2 [33], final concentration 10 nM each. RNF8 was depleted with RNF8 specific siRNA according to Ismail, et al. [32], final concentration 100 nM.

To determine the clonogenic survival of HeLa.S-Fucci G1 or G2 cells, cells were detached immediately after irradiation and flow-cytometry sorted for G1 and G2 phase based on the Fucci system. G1- and G2-phase cells were then seeded for a clonogenic survival assay as described earlier and incubated for 12 days before colonies of 50 or more cells were quantified [41]. Data points were fitted to a linear-quadratic survival model.

### 2.2. Fluorescence-Activated Cell Sorting

To perform cell-cycle specific analyses, we used in part of our experiments HeLa.S-Fucci cells. Based on the Fucci (fluorescence ubiquitination cell-cycle indicator) system we sorted detached asynchronous HeLa.S-Fucci cells for G1 and G2 phase cells with an S3e Cell Sorter (Bio Rad, Hercules, CA, USA). The proper sorting was frequently tested by western-blot analysis of G2-phase specific cyclin A (Appendix A).

### 2.3. Irradiation

Cells were irradiated with X-rays (X-ray tube MXR 320-26, Seifert/GE, Germany; 250 keV, 16 mA; LET 2 keV/µm), α particles (track-averaged LET approximately 153 keV/µm), or heavy ions. α-particles irradiation was performed with an ^241^Am source according to Maier*,* et al. [42]. Heavy-ion irradiation took place at the GSI Helmholtz Center for Heavy Ion Research (Darmstadt, Germany). Heavy-ion irradiation of cell cultures intended for immunofluorescence microscopy of RPA, CtIP, or Ku80 was performed in such a way that the ions hit the cell monolayer in a low angle. Thus, upon immunostaining of DSB-relevant factors, ion-induced DNA damage became visible along the ions’ trajectories in a streak-like pattern [43]. For DSB-repair kinetics, cells were irradiated perpendicularly. Accelerator access and the selection of ions available is limited; hence some ion data are from single experiments (as indicated). At the UNILAC linear accelerator experiments with low-energy carbon ions at a primary energy of 6.5 MeV/nucleon (LET 325 keV/µm) or 11.4 MeV/nucleon (LET 168 keV/µm), or low-energy iron ions at a primary energy of 11.4 MeV/nucleon (LET 2875 keV/µm) were performed. Irradiation with high-energy iron ions took place at the SIS beam line at a primary energy of 1 GeV/nucleon (LET 155 keV/µm). Several data were repeated with different but comparable ion species within one experiment upon availability at the accelerator.

### 2.4. Immunostaining and Confocal Microscopy

For immunostaining, cells were generally fixed in 2% formaldehyde and permeabilized as described earlier [13]. For immunostaining of Ku80 we followed the protocol of Britton, et al. [44] with some changes: Pre-extraction with CSK+R buffer was performed 3x instead of 2 × 3 min, fixation occurred for 15 min with 4% instead of 2% formaldehyde, and permeabilization was done for 10 min instead of 5 min with PBS, 0.2% Triton X-100. Primary antibodies for immunofluorescence staining were diluted in 1x PBS, 0.4% BSA: α53BP1 (Ab-1) (rabbit, Calbiochem, PC712, 1:500), αCENP-F (rabbit, Novus, NB500-101, 1:750), αCtIP (rabbit, Bethyl Laboratories, A300-488A, 1:100), αGeminin (mouse, clone 1A8, Novus Biologicals, H00051053-M01, 1:100), αγH2AX (Ser 139) (mouse, clone JBW301, Millipore, 05-636, 1:500), αKu80 (mouse, clone 111, Thermo Scientific, MA5-12933, 1:100), αRPA/P34 (mouse, clone 9H8, Sigma, R1280, 1:3000). To visualize the primary antibodies, secondary Alexa 488-, Alexa 568-, or Alexa 647-conjugated αmouse or αrabbit antibodies from donkey or goat were used (Invitrogen, Life Technologies, Sigma-Aldrich). DNA was counterstained with DAPI (AppliChem; 1 µg/mL). In the end, the samples were mounted in Slow Fade Diamond Antifade (Invitrogen, Waltham, MA, USA) and analyzed at a Leica TCS or Nikon spinning disk microscope.

### 2.5. EdU Labeling

For cell-cycle specific analysis of DSB-repair kinetics we added EdU (10 µM final concentration) immediately after irradiation. As this thymidine analogue is integrated during replication, it allows distinguishing G2 cells that were irradiated in G2 phase from G2 cells that were irradiated in G1 or S phase and went through the cell cycle into G2. To visualize EdU we used the EdU Click-647 Kit from Roth (Art. No. 7777.1).

### 2.6. Cell-Cycle Specific DSB-Repair Kinetics

To determine repair kinetics of radiation-induced DSBs we used a γH2AX-foci analysis according to Löbrich*,* et al. [45]. γH2AX foci were quantified at different time points after irradiation. γH2AX-foci numbers of non-irradiated cells were subtracted and the data were normalized to data 15 min after irradiation, which were considered 100% DSB damage upon irradiation. Thus, the resulting ratio represents the fraction of remaining DSBs after irradiation. The absolute number of γH2AX foci 15 min after irradiation was similar between the tested genotypes. The cell-cycle phase of each cell was determined based on its nuclear DAPI signal with the software MetaCyte (MetaSystems, Heidelberg, Germany). Adding EdU after irradiation allowed distinguishing G2 cells that were irradiated in G2 phase from G2 cells that were irradiated in a previous cell-cycle phase. Only G2 cells that were irradiated in G2 phase were analyzed. Using Metafer 4 (MetaSystems), images of the DNA, EdU, and γH2AX signal were taken and analyzed automatically resulting in DSB-repair kinetics. Per time point and cell-cycle phase, approximately 300 cells on average were analyzed.

### 2.7. Immunoprecipitation (IP) and Pull down (PD)

Cells were detached and rinsed in PBS before lysed in IP lysis buffer (20 mM Tris-HCl pH 8.2, 150 mM NaCl, 1% Triton-X-100, 5 mM sodium butyrate, 50 µM PR619, 5 mM 1,10-Phenanthrolin, protease and phosphatase inhibitors). Per IP or PD sample, 10^6^ cells were lysed in 200 µL buffer and incubated for 5 min on ice. Afterwards, the samples were sonicated in order to fractionate DNA.

For CtIP, IP 10 µg αCtIP (rabbit, Bethyl Laboratories, Montgomery, TX, USA, A300-488A) antibody was coupled to Protein G Dynabeads (Thermo Fisher, Waltham, MA, USA), 160 µL cell lysate added, and incubated at 4 °C for 3 h while rotating. For PD of ubiquitinated proteins TUBE2 magnetic beads (UM402M, LifeSensors, Malvern, PA, USA) were incubated with 160 µL cell lysate at 4 °C overnight while rotating. After several washing steps in PBS, 0.1% BSA, the captured proteins were detached from the beads by boiling in 30 µL Lämmli lysis buffer; 15 µL of this extract was used per western- or ligand-blot analysis.

### 2.8. Western-Blot Analysis and Ligand-Blot Analysis

Cell, immunoprecipitation, or pull-down extracts were separated in Mini-PROTEAN^®^ TGX™ (Bio Rad) gels, transferred onto nitrocellulose membrane (Bio Rad) and immunoblotted with αBRCA1 (rabbit, Santa Cruz, sc-642, 1:50); αCtIP (rabbit, Bethyl Laboratories, A300-488A, 1:100); αKu80 (mouse, clone 111, Thermo Scientific, Waltham, MA, USA, MA5-12933, 1:200); αRIF1 (rabbit, Bethyl Laboratories, A300-569A, 1:1000); αRNF8 (rabbit, Millipore, Burlington, MA, USA, 09-813, 1:1000); αRNF138 (rabbit, Sigma, St. Louis, MO, USA, SAB4502131, 1:1000); αRNF138, biotin conjucated (rabbit, AntibodyGenie, PACO0015142, 1:250); or αvinculin (mouse, Abcam, Cambridge, UK, ab18058, 1:5000). As secondary antibodies served Poly-HRP-α-mouse (goat, Thermo Scientific, 1:500) or Poly-HRP-α-rabbit (goat, Thermo Scientific, 1:500). In ligand blots, ubiquitin was detected with TUBE2-Biotin (UM302, LifeSensors) and Streptavidin-HRP (Thermo Scientific, N100, 1:500). To visualize proteins of interest, Pierce ECL2 western blotting substrate (80196) was used.

### 2.9. Statistical Analyses

For *t*-test analyses GraphPad Prism 8 (GraphPad Software, San Diego, CA, USA) or Excel 2016 (Microsoft Office, Redmond, WA, USA) was used. Mann-Whitney analyses were performed with GraphPad Prism 8. The sign test was performed without specific statistical software. The significance level for all tests was 0.05.

## 3. Results

### 3.1. BRCA1 and RIF1 Play No Major Role in Regulating Resection of Complex DSBs in G1 Cells

A prominent member of resection regulation is BRCA1, which in G2 cells together with the resection factor CtIP overcomes the resection block mediated by 53BP1 and its binding partner RIF1 [23]. As we found earlier that CtIP is important for DSB resection in G1 cells [19], we studied whether its partner BRCA1 is required to permit resection in G1 phase. To quantify resection-positive cells, we used immunofluorescence staining of the single-strand DNA binding protein RPA that serves as resection marker [46] and the cell-cycle marker CENP-F [47] (Figure 1b). To facilitate distinguishing RPA-positive cells, we irradiated the cells in a low angle (Figure 1a) causing DNA damage in a streak-like pattern. As expected, in S/G2-phase cells, the depletion of BRCA1 by RNAi significantly decreased the fraction of RPA positive, i.e., resection positive cells, upon induction of complex DNA damage with carbon-ion irradiation (Figure 1c). Yet, under these conditions, BRCA1 depletion had no influence on the fraction of resection positive G1-phase cells. Furthermore, depleting RIF1, which acts opposing BRCA1′s role in resection regulation, had also no effect on the fraction of resection-positive cells after carbon-ion irradiation (Figure 1d). These data suggest that BRCA1 and RIF1 are no major players in controlling the resection block of complex heavy-ion-induced DSBs in G1-phase cells.

### 3.2. The E3 Ubiquitin Ligases RNF8 and RNF138 Are Required for Radiation-Dependent Ubiquitination in G1 Cells

Ubiquitination plays a crucial role in regulating DSB resection in G2-phase cells by affecting several steps of resection [27]. Here we studied whether irradiation with ionizing radiation of different quality can induce ubiquitination in G1-phase cells (Figure 2). To quantify ubiquitinated proteins specifically in this cell cycle phase, we used the HeLa.S-Fucci cell system that allows for cell-cycle specific flow-cytometric sorting of living cells [39].

Ubiquitinated proteins of irradiated G1-phase cells were pulled down 1 h after exposure and then detected with Tandem Ubiquitin Binding Entities (TUBE) for quantification. We induced DNA damage with X-ray (low LET) or iron-ion (high LET) irradiation. 1 h after irradiation with 30 Gy X-rays we observed a marginal increase in ubiquitinated proteins. Notably, following 30 Gy iron-ion irradiation, which causes more complex DNA damage than X-rays, we detected a drastic increase in ubiquitinated proteins. This radiation-dependent increase of ubiquitinated proteins was not observed when RNF8 or/and RNF138, E3 ubiquitin ligases known to participate in resection regulation prior to HR, were depleted (Figure 2). Interestingly, a lack of RNF138 causes an increase of ubiquitinated proteins that is no longer apparent upon irradiation and is not observed when RNF8 is additionally depleted. Due to limited access to the heavy-ion beamline this experiment was performed only once, thus further experiments are required to corroborate these observations. Overall, our data suggest that RNF8 and RNF138 are required for radiation- related ubiquitination in G1-phase cells and prompted us to study their relevance for resection of complex DSBs in G1 cells.

### 3.3. RNF138 Is Involved in Resection of Complex DSBs in G1-Phase Cells

To analyze whether the function of RNF8 and RNF138 is necessary for the resection of complex, radiation-induced DSBs in G1-phase, we induced such damage with α-particles, carbon ions, or iron ions. All three types of radiation induce a substantial fraction of resected DSBs [19], which is mirrored in the fraction of cells that are positive for the resection marker RPA upon irradiation (Figure 3).

RNF8 depletion had no influence on the fraction of RPA positive cells upon induction of complex DSBs with α-particles (Figure 3a). Since depletion of RNF8 also decreases the recruitment of the resection antagonist 53BP1 [48] (Appendix A), it is well conceivable that this counteracts the resection-impairing effect of RNF8 depletion and thus, RNF8′s function in resection is not easily studied. Depletion of RNF138 by knockout of its gene in two independent cell lines, however, showed a clear decrease of resection-positive cells in G1 phase after induction of complex DNA damage with carbon- or iron-ion irradiation (Figure 3b). From this, we conclude that in G1 phase RNF138 is necessary for resection processes.

### 3.4. RNF138 Is Essential for DSB Recruitment of the Resection Factor CtIP in G1-Phase Cells

Due to the observed effect on resection in G1 cells by RNF138 depletion and since the resection factor CtIP requires RNF138-dependent ubiquitination for its DSB recruitment within HR repair [33], we investigated the recruitment of CtIP to heavy-ion-induced DSBs in dependence of RNF138 and the cell cycle in three different cell lines depleted for RNF138 by knockdown or knockout of its gene (Figure 4). In all analyzed experimental conditions, RNF138 depletion clearly caused a decrease of CtIP-positive G1 cells upon carbon- or iron-ion irradiation. In the case of NFFhTERT cells, the effect was less strong, most likely since the RNF138 knockdown was not complete ( Appendix A). Taken together, as in G2-phase cells, in G1-phase cells the recruitment of the resection factor CtIP to complex DSBs requires RNF138.

### 3.5. RNF138 Interacts with CtIP and Is Required for CtIP’s Ubiquitination in G1-Phase Cells

As we saw that the recruitment of CtIP to complex DSBs in G1-phase cells relies on RNF138, we analyzed whether this dependence is based on a direct interaction of CtIP and RNF138 in G1 cells (Figure 5). Therefore, we immunoprecipitated CtIP of G1 cells (HeLa.S-Fucci) irradiated with carbon ions and performed a western-blot analysis detecting RNF138 (Figure 5a). To allow the comparison to conditions with less complex DNA damage, we also analyzed cells irradiated with X-rays. We found that for all conditions tested, CtIP co-immunoprecipitated RNF138. These data support the idea that in G1 cells RNF138 interacts directly with CtIP in order to ubiquitinate it. However, the interaction of CtIP and RNF138 in G1 cells is not dependent on irradiation. This conforms to earlier observations in populations of mixed cell-cycle distribution [33].

In a further step, we studied whether RNF138 promotes CtIP’s ubiquitination in G1 cells in a radiation-dependent manner (Figure 5b). We irradiated cells with iron ions or X-rays. Although the latter radiation quality showed no influence on the fraction of ubiquitinated proteins in the absence of RNF138 (Figure 2), we considered that on the level of single proteins it may well show an effect. Thus, we pulled down ubiquitinated proteins of X-ray, iron-ion, or non-irradiated G1 cells either proficient or deficient for RNF138 and analyzed whether CtIP was among the pulled-down proteins. We found that upon irradiation the fraction of ubiquitinated CtIP is increased, indicating that CtIP becomes irradiation-dependent ubiquitinated. As after iron-ion irradiation more CtIP was found to be ubiquitinated than upon X-ray irradiation, the data suggest that with increasing DNA damage complexity CtIP becomes ubiquitinated to a greater extent. Furthermore, in RNF138 knockout cells, hardly any CtIP was pulled down from three independent G1-cell extracts (Figure 5b, lanes 5–7) indicating that RNF138 is the main if not sole E3 ubiquitin ligase responsible for this ubiquitination. Thus, we conclude that in G1-phase cells RNF138 ubiquitinates CtIP upon induction of DSBs to promote its DSB accrual.

### 3.6. Removal of Ku80 in the Context of DSB Resection Does Not Require RNF138 in G1-Phase Cells

In G2 cells prior to HR, RNF138 promotes resection not only via the regulation of CtIP’s DSB recruitment, but also via the removal of the resection antagonist Ku80 from DSB sites [49]. In order to find out whether resection of complex DSBs in G1-phase cells is facilitated by RNF138-dependent Ku80 removal, we studied the presence of Ku80 at heavy-ion-induced DSBs 15 min and 1 h after irradiation in dependence of RNF138 (Figure 6). 15 min after angular irradiation with carbon or iron ions we detected Ku80 at DSBs along ion traversals in a majority of irradiated, asynchronous cells both in RNF138 proficient and deficient cells. Notably, 1 h after irradiation with both types of ions Ku80 was hardly detectable independent of the presence of RNF138; the fraction of Ku80-positive cells decreased by about 90%. The similar behavior of wild-type and RNF138-depleted cells suggests that Ku80 removal from complex heavy-ion-induced DSBs does not require RNF138 in G1 cells.

### 3.7. RNF138 Deficiency Diminishes DSB Repair and Survival of Heavy-Ion Irradiated Cells

Since RNF138′s function in resection seems to rely mainly on CtIP, we wanted to explore whether RNF138 is important for repair of complex DSBs in G1 cells, as previously described for CtIP [19]. Hence, we analyzed in a γH2AX-foci assay the repair of radiation-induced DSBs in RNF138 proficient (wt) and deficient (RNF138 KO) cells (Figure 7a). We found that upon X-ray irradiation, DSB repair in G1 cells was not influenced by RNF138 depletion. Yet, upon induction of more complex DNA-damage with carbon ions, DSBs induced in RNF138-depleted cells were less well rejoined in G1-phase cells; 10 h after irradiation, RNF138-deficient G1 cells have about 30% more remaining DSBs than wild-type cells. These repair data mirror those obtained earlier in cells depleted for CtIP [19]. RNF138-deficient G2 cells showed diminished repair both after X-ray and ion irradiation. For the latter radiation quality, an impairment was detectable even though in the time window starting 4 h post carbon-ion irradiation DSB repair was practically not detected in G2 cells.

Interestingly, similar to a depletion of CtIP [50], also a depletion of RNF138 was found to rescue an Artemis repair deficiency of X-ray-induced DSBs (Figure 7b); this indicates that RNF138 participates in the repair of at least a subset of X-ray-induced DSBs. At α-particle-induced complex DSBs, however, depletion of RNF138 cannot rescue an Artemis repair defect (Appendix A).

The relevance of RNF138 for DSB repair in G1-phase cells and thus survival was further explored in clonogenic survival assays of G1 cells irradiated with X-rays or carbon ions in dependence of RNF138 (Figure 7c). As expected, due to the increased DNA damage complexity, carbon-ion-irradiated G1 cells survived less well than X-ray-irradiated G1 cells. Notably, while RNF138 deficiency did not affect survival of G1 cells upon X-ray irradiation, it decreased the survival of G1 cells upon carbon-ion irradiation significantly (paired sign test; *p* = 0.031). In G2 cells, RNF138 deficiency diminished the ability to survive X-ray and carbon irradiation. Here, only the X-ray data proofed to be significant (paired sign test; *p* = 0.031), yet for the carbon data the tendency is clearly visible. Due to limited access to the beamline, the carbon-ion experiment was performed only once. Nevertheless, the observed survival data are in line with the DSB-repair data (Figure 7a). Taken together, the DSB repair and clonogenic survival data suggest that RNF138′s function is important in G2 phase, which conforms to earlier observations [32]. The data further suggest that with increasing DNA-damage complexity, RNF138 becomes increasingly relevant for DNA repair and radiation survival in all cell-cycle phases.

## 4. Discussion

Radiation-induced complex DNA damage, particularly clustered DSBs as induced by densely ionizing heavy ions, pose serious challenges to the cellular repair machinery [13,14,51]. Earlier studies revealed that the break ends of heavy-ion inflicted DSBs are in all cell-cycle phases frequently processed by resection [19,20]. If not repaired by HR, a pathway that occurs in S and G2 phase only, this processing causes inevitably loss of DNA and thus genetic information [2,21]. Hence, break-end processing by resection in G1-phase cells results in erroneous DSB repair and adds to the biological effectiveness of heavy ions. The present study is aimed at revealing the processes that lead to DSB resection in the G1 cell-cycle phase.

Here we found that neither depleting RIF1 nor BRCA1–two antagonists within the control of DSB resection via the 53BP1/RIF1 resection block [52]–had a visible effect on resection of complex heavy-ion-induced DSBs in G1 cells 1 h after irradiation (Figure 1). Our data suggest that this mechanism of DSB-resection control is not a master regulator but rather one of several layers of resection control. Since DSB resection in G1 cells seems to always compromise the quality of DSB repair and hence needs to be tightly controlled, multiple layers of regulation could be expected. A previous report indicating less resection-positive alpha-particle-induced DSBs upon BRCA1 depletion [50] may be due to the later observation time post irradiation (6 h). At that time, the lack of BRCA1 in its function to accelerate CtIP-mediated resection may decrease the detection of resection-positive DSBs [53]. Unlike the depletion of BRCA1, depletion of CtIP–BRCA1′s partner in abolishing the 53BP1/RIF1 resection block in S/G2 phase–showed a severe effect on DSB resection in heavy-ion-irradiated cells as previously described [19,20]. This most likely owes to CtIP’s functions in the resection process itself aside from its function in resection regulation.

As DSB processing by resection has serious consequences, it is controlled by several mechanisms, among them ubiquitination [27]. We found that ionizing radiation increases the fraction of ubiquitinated proteins in G1-phase cells (Figure 2). Furthermore, our data suggest that upon induction of more complex DNA damage an increased fraction of proteins becomes ubiquitinated in this cell-cycle phase. Although this observation needs to be corroborated by additional experiments, it is supported by earlier data reported upon X-ray and carbon-ion irradiation of a cell population of mixed cell-cycle stage [54]. Importantly, our data suggest that the radiation-dependent ubiquitination in G1 phase relies on RNF8 and RNF138 (Figure 2), two E3 ubiquitin ligases required for resection control in the context of HR repair [32,35]. Furthermore, in the absence of DNA damage the function of these enzymes might be interrelated, as suggested by the increase in overall ubiqutination in RNF138KO cells that is not observed when RNF8 is additionally depleted.

Depleting RNF8 had no influence on the fraction of resection-positive cells, neither in G2 nor in G1 phase. However, RNF8 depletion decreases the recruitment of the resection antagonist 53BP1 (Appendix A), whose accumulation at DSBs relies on RNF8-dependent histone H1 and L3MBTL2 ubiquitination [55,56]. Therefore, we cannot rule out that a resection-impairing effect of RNF8 depletion is in part counteracted by concomitantly decreased 53BP1 recruitment at DSBs.

Depleting RNF138 on the other hand resulted in less resection positive G1 cells upon induction of DSBs by carbon or iron ions (Figure 3b), which supports the idea that RNF138 is involved in resection processes in G1 cells. Promotion of resection of complex DSBs in G1 cells seems not to rely on RNF138-dependent removal of Ku80 from the break site, since RNF138 depletion did not increase the retention of Ku80 at carbon- or iron-ion-induced DSBs (Figure 6). Rather, in G1 cells RNF138 facilitates DSB resection by promoting the recruitment of the resection factor CtIP to complex DSBs. This is demonstrated by our observation that RNF138 deficiency causes a severe decrease in cells depicting CtIP foci upon induction of complex DSBs (Figure 4). Considering earlier data by Schmidt, et al. [33], our data (Figure 4 and Figure 5) support a model in which CtIP recruitment to DSBs in G1 cells depends on its radiation-dependent ubiquitination by RNF138. Since the RNF138-CtIP interaction is not radiation-dependent itself (Figure 5a), one or more factors should control the radiation-dependent ubiquitination of CtIP by RNF138.

Interestingly, a lack of RNF138 had a far stronger effect on the DSB recruitment of CtIP than on the fraction of resection positive cells (compare Figure 3b and Figure 4). As CtIP is not the sole resection factor but resection also involves additional factors, i. e. MRE11, EXO1, and DNA2/BLM, these factors may maintain resection in the absence of CtIP as it was seen earlier [57,58]. This may occur especially in G2, where RNF138 depletion had only a minor effect. The strong effect an RNF138 knockout has on CtIP’s recruitment emphasizes the exclusive role of this E3 ubiquitin ligase for CtIP’s ubiquitination that is required for its recruitment to DSBs. This is also indicated in the lack of CtIP in the pool of ubiquitinated proteins of G1 cells when RNF138 is missing (Figure 5b).

DSB-repair assays (Figure 7a) further suggest that RNF138′s role in repair of complex DSBs induced in G1 phase is mainly based on its function on CtIP, since the effect of RNF138 depletion on DSB repair is comparable to the effect that CtIP depletion had on DSB repair (Figure 7a and [19]). An additional result, which supports this notion, is that depleting RNF138 rescues an Artemis defect in G1 cells (Figure 7b) comparable to CtIP depletion [50]. This result revealed further that RNF138 participates in the repair of at least a subset of X-ray-induced DSBs in G1 cells, which could not be concluded from DSB repair and survival data upon X-ray-irradiation. RNF138 seems to be dispensable for the repair of X-ray induced DSBs in G1 cells since its deficiency, which acts upstream of Artemis in resection-dependent repair, allows to circumvent an Artemis requirement for DSB repair (Figure 7b). Yet, in the case of complex DSBs RNF138 appears to be important for their repair, (Appendix A). At this type of DNA damage, resection seems vital for repair in G1 cells, as depletion of RNF138 can no longer rescue a deficiency of Artemis. This view is supported by our findings that RNF138-deficient G1 cells repair and survive less well carbon-ion-induced DSBs in comparison to X-ray-induced DNA damage (Figure 7a,c). Taken together, these results mirror those of CtIP depleted, Artemis-deficient cells [50] and thus, further support the idea that RNF138′s function in G1 phase is mainly via its binding partner and ubiquitination target CtIP.

The reduced repair of complex DSBs observed in cells missing RNF138 is also manifested in their clonogenic survival, showing slightly reduced survival of both G1- and G2-phase cells (Figure 7c). Cell survival was overall lower upon carbon-ion irradiation due to the increased ionization density of this radiation type. Repair of carbon-ion-induced DSBs in wild-type cells was less efficient in G2 than in G1-phase cells, yet their surviving fractions are similar. This finding may be a consequence of the different time windows of observation and hence opportunity of DSB repair (hours versus days) and a consequence of DSB-repair quality, which is comprised in survival data but not repair data revealed by γH2AX-foci assays.

We aimed to elucidate the regulation of DSB resection that occurs at heavy-ion-induced complex DSBs in G1-phase cells. Our data suggest that RIF1 together with its partner 53BP1 is no major obstacle for resection initiation at that type of DNA damage. Here we show that the E3 ubiquitin ligase RNF138 functions in resection of DSBs in G1 phase. This function becomes increasingly important with increasing DNA-damage complexity and consists of the radiation-dependent ubiquitination of the resection factor CtIP to facilitate its recruitment to DSBs; a model summarizing these findings is depicted in the graphical abstract. Thus, we have revealed a layer of regulation of DSB processing in dependence of DNA-damage complexity, which may be exploited to tailor heavy-ion radiotherapy.

## Figures and Tables

**Figure 1 cells-11-02561-f001:**
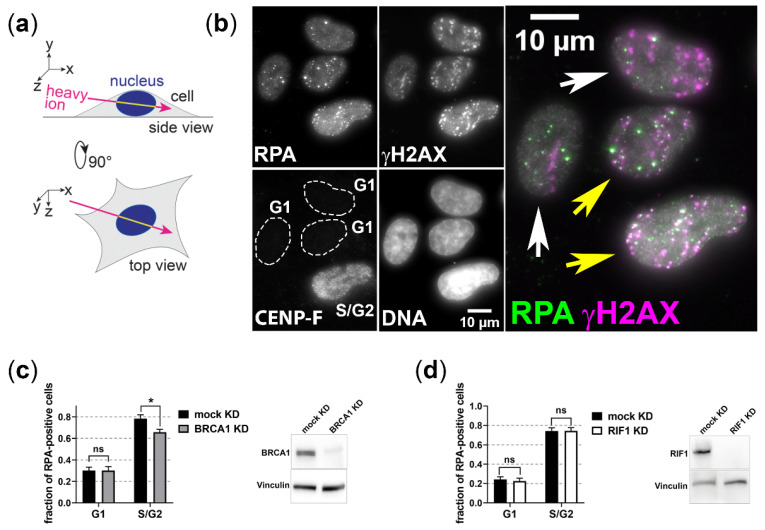
**BRCA1 and RIF1 do not modulate resection of complex DSBs induced in G1-phase cells.** U2OS cells were depleted for BRCA1 (BRCA1 KD) or RIF1 (RIF1 KD) by RNAi or mock depleted and irradiated with 5 × 10^6^ carbon ions/cm^2^ (6.5 MeV/nucleon) in a low angle. The depletion was proven by western-blot analysis. 1 h after irradiation cells were fixed for immunofluorescence staining of the resection marker RPA, the DSB marker γH2AX, and the cell-cycle marker CENP-F (CENP-F positive: S, G2, and M phase; CENP-F negative: G1 phase). DNA was visualized by DAPI staining, which allowed distinguishing M-phase cells. Irradiated cells (i.e., cells with streak-like γH2AX signal) that show colocalization of RPA-with γH2AX along the ion track were counted RPA positive (yellow arrows); those that show no colocalization of RPA and γH2AX were counted RPA negative (white arrows). The experiments were performed twice and at least 100 cells per condition were analyzed. Error bars: SEM. Two-tailed Mann-Whitney test, *p* values: ns, not significant; * *p* < 0.05. (**a**) Scheme of angular heavy-ion irradiation of culture cells, which causes streak-like patterns of ion traversals. (**b**) Representative image of carbon-ion-induced RPA signal in G1 and S/G2 cells. (**c**) Quantitative analysis of irradiated RPA-positive G1 and S/G2 cells depleted or mock depleted for BRCA1. (**d**) Quantitative analysis of irradiated RPA-positive G1 and S/G2 cells depleted or mock depleted for RIF1.

**Figure 2 cells-11-02561-f002:**
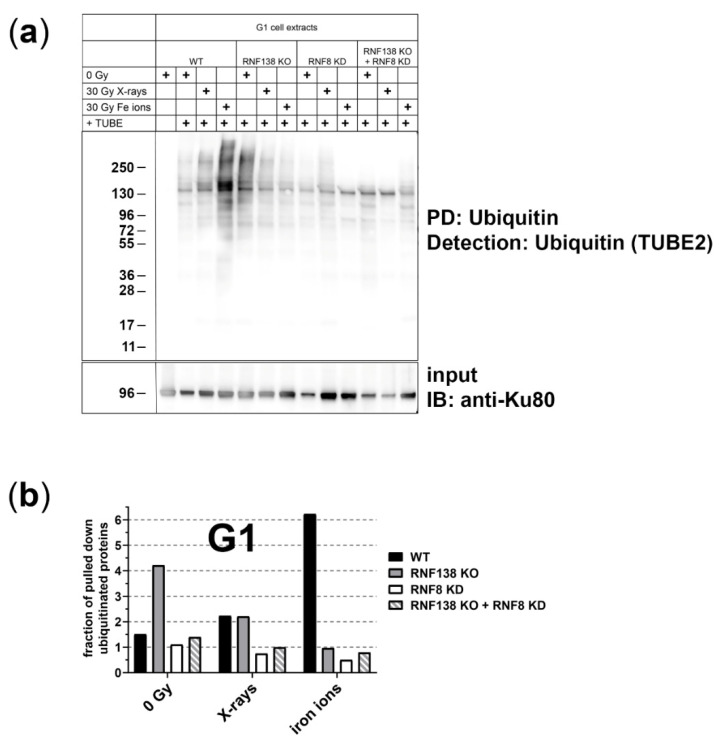
**Protein-pulldown experiments point out that the E3 ubiquitin ligases RNF8 and RNF138 are required for radiation-related ubiquitination in G1 cells**. (**a**) HeLa.S-Fucci cells proficient (wt) or deficient for RNF138 (RNF138 KO) and/or RNF8 (RNF8 KD) were irradiated either with 30 Gy X-rays or 30 Gy high-energy iron ions. 1 h after irradiation G1 cells were isolated via flow cytometry, their proteins extracted, and ubiquitinated proteins pulled down and visualized with TUBE2. PD: pull down; IB: immunoblot. (**b**) For quantification of ubiquitinated proteins, western-blot detection of Ku80 in the input-protein extracts (whole-cell extracts that were used for PD, see (**a**) was used for normalization to equal loading. For all conditions, the extracts for the pull-down reactions were performed from 10^6^ cells each. Differences in the protein concentration of these extracts are based on the success of cell lysis and reflected in the Ku80 signal. *n* = 1.

**Figure 3 cells-11-02561-f003:**
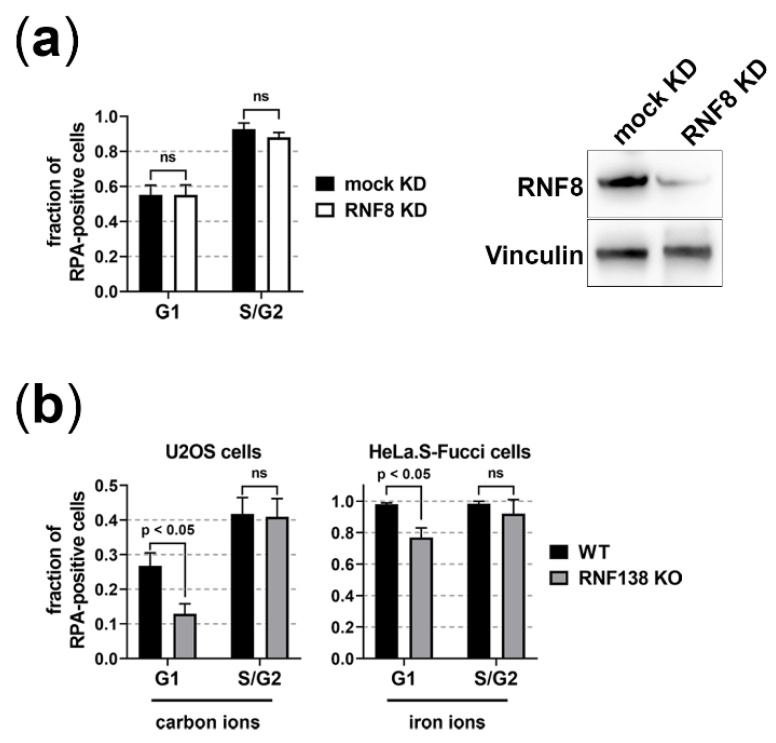
**RNF138 is involved in resection of complex DSBs in G1-phase cells.** To visualize resection, we fixed the cells 1 h after irradiation and immunostained the resection marker RPA. To allow cell-cycle specific analyses, we used the endogenous Fucci system in HeLa.S-Fucci cells. In NFFhTERT and U2OS cells, we immunofluorescence stained the cell-cycle marker CENP-F (CENP-F positive: S, G2, and M phase; CENP-F negative: G1 phase). DNA was stained with DAPI, to visualize the nuclei of all cells and to distinguish M-phase cells. At least 100 cells per condition were analyzed from at least 6 fields of view in order to average the number of RPA positive cells. *n* = 1, error = SEM for the different fields of view. Two-tailed Mann-Whitney test, *p* values < 0.05 were considered significant; ns, not significant. (**a**) NFFhTERT cells were depleted or mock depleted for RNF8 by RNAi. In addition to a western-blot analysis, the RNF8 depletion was verified by quantification of radiation-induced 53BP1 foci per nucleus (Appendix A) and irradiated with 2 Gy α-particles to induce complex DNA damage. Cells showing more than 5 RPA foci (i.e., RPA signal well above background) after irradiation were considered resection positive. The average number of RPA positive non-irradiated cells was subtracted. (**b**) U2OS or HeLa.S-Fucci cells RNF138 proficient or deficient (wt or RNF138 KO) were irradiated in a low angle (5 × 10^6^ p./cm^2^) with carbon ions (6.5 MeV/nucleon) or low-energy iron ions. Irradiated cells (i.e., cells showing γH2AX streaks) positive for RPA-decorated ion traversals within their nuclei 1 h after irradiation were considered resection positive. The RNF138 deficiency caused by an RNF138 knockout (KO) was checked by western-blot analysis (supplemental Appendix A).

**Figure 4 cells-11-02561-f004:**
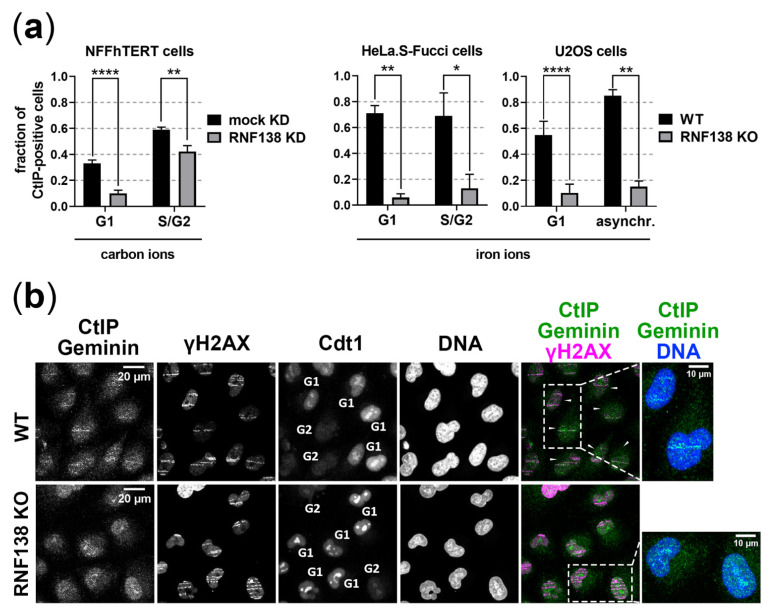
**RNF138 is required for the recruitment of CtIP to DSBs in G1-phase cells.** NFFhTERT, HeLa.S-Fucci, or U2OS cells, the latter ectopically expressing CtIP-mCherry, RNF138 proficient or deficient [mock KD and RNF138 KD (see Appendix A) or wt and RNF138 KO, respectively] were irradiated in a low angle (5 × 10^6^ p./cm^2^) with carbon ions (6.5 MeV/nucleon) or low-energy iron ions and fixed 1 h after irradiation. The samples were immunofluorescence-stained for CtIP and the cell-cycle marker Geminin (Geminin positive: S and G2 cells, Geminin negative: G1 cells; NFFhTERT and U2OS only) or for CtIP and the DSB marker γH2AX (HeLa.S-Fucci cells). DNA was counterstained with DAPI to visualize the nuclei of all cells. Cells positive for CtIP-decorated ion traversals (streak-like pattern) within their nuclei were considered CtIP positive. (**a**) At least 100 cells per condition were analyzed from at least 6 fields of view in order to average the number of CtIP-positive cells. *n* = 1, error = SEM. Two-tailed Mann-Whitney test, *p* values: * *p* < 0.05; ** *p* < 0.01; **** *p* < 0.0001. (**b**) Representative image of CtIP signal in WT and RNF138 KO HeLa.S-Fucci cells. The CtIP signal is overlaid by the signal of mAG-tagged Geminin, a part of the Fucci system. mKO2-tagged Cdt1 of the Fucci system served as cell-cycle marker. White arrows indicate CtIP-positive cells. An enlarged version of the areas highlighted by dashed boxes is shown to the right.

**Figure 5 cells-11-02561-f005:**
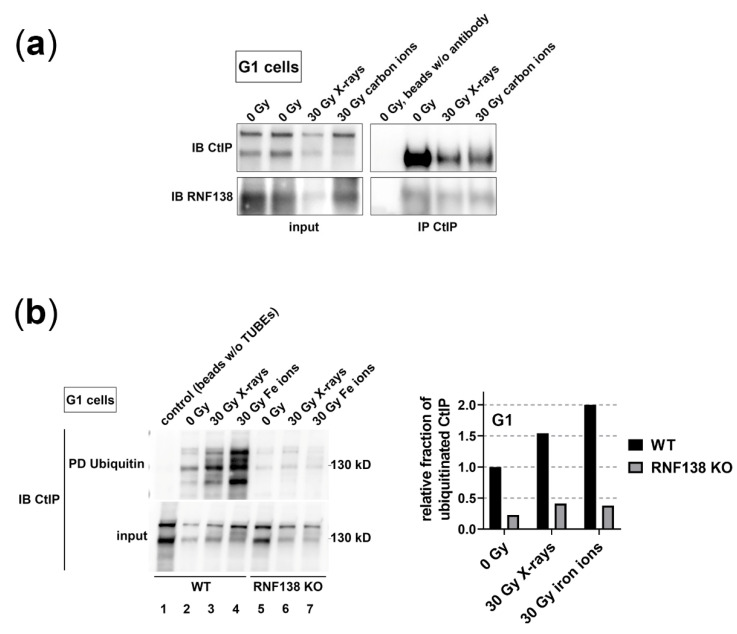
**RNF138 is required for CtIP’s ubiquitination in G1-phase cells.** (**a**) HeLa.S-Fucci cells were irradiated either with 30 Gy X-rays or 30 Gy carbon ions (6.5 MeV/nucleon). 1 h after irradiation, G1 cells were isolated via flow cytometry. For each condition, proteins were isolated from 10^6^ G1 cells and CtIP immunoprecipitated. A western-blot analysis revealed that CtIP was successfully precipitated (IP) from the extracts (input) and that RNF138 was co-immunoprecipitated. IB: immunoblot (**b**) HeLa.S-Fucci cells proficient (wt) or deficient for RNF138 (RNF138 KO) were irradiated either with 30 Gy X-rays or 30 Gy high-energy iron ions. 1 h after irradiation G1 cells were isolated via flow cytometry and ubiquitinated proteins pulled down with TUBE from whole-cell extracts (“input”) (left). CtIP was detected in a pull down (PD) of ubiquitinated proteins by means of western-blot analysis. A PD reaction with cell extract of non-irradiated cells and magnetic beads without TUBE2 attached served as a PD control (lane 1). The signals of ubiquitinated proteins were quantified with ImageJ (right). IB: immunoblot.

**Figure 6 cells-11-02561-f006:**
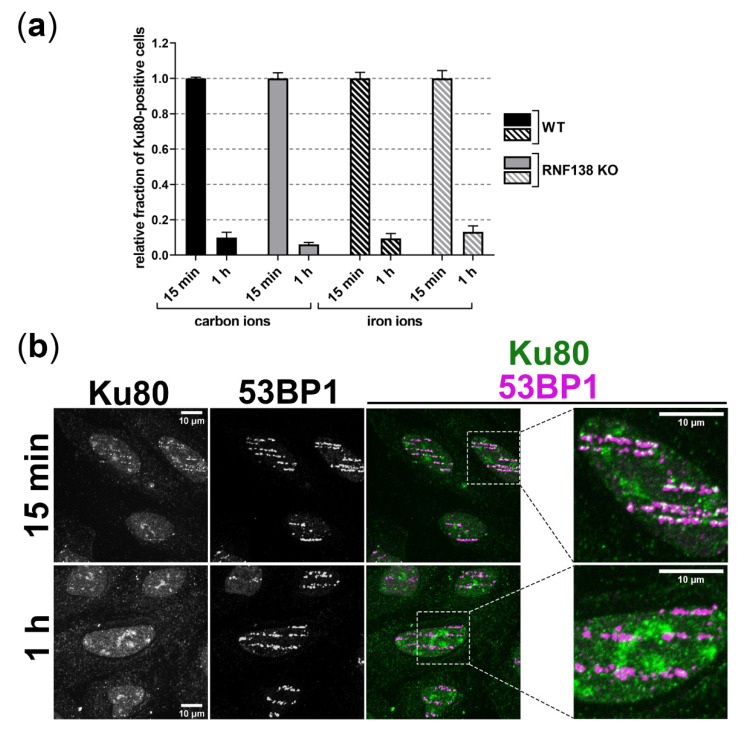
**Ku80 vacates complex heavy-ion-induced DSBs independent of RNF138.** U2OS cells proficient (wt) or deficient for RNF138 (RNF138 KO) were irradiated in a low angle (5 × 10^6^ p./cm^2^) with low-energy carbon (11.4 MeV/nucleon) or high-energy iron ions and fixed 15 min or 1 h after irradiation. Ku80 and 53BP1 (DSB marker) were immunofluorescence stained. Ku80-positive, irradiated (53BP1-positive) cells were quantified. Per sample, at least 100 irradiated cells were analyzed and 6–10 fields of view were enumerated in order to average the number of Ku80-positive cells. Within the irradiated area, all cells were hit by ions. *n* = 1, error = SEM (**a**) Fraction of Ku80-positive cells 15 min or 1 h after irradiation in dependence of RNF138. (**b**) Exemplary images of iron-ion-irradiated wt cells 15 min and 1 h after irradiation and immunofluorescence staining of Ku80 and 53BP1. An enlarged version of the areas highlighted by dashed boxes is shown to the right.

**Figure 7 cells-11-02561-f007:**
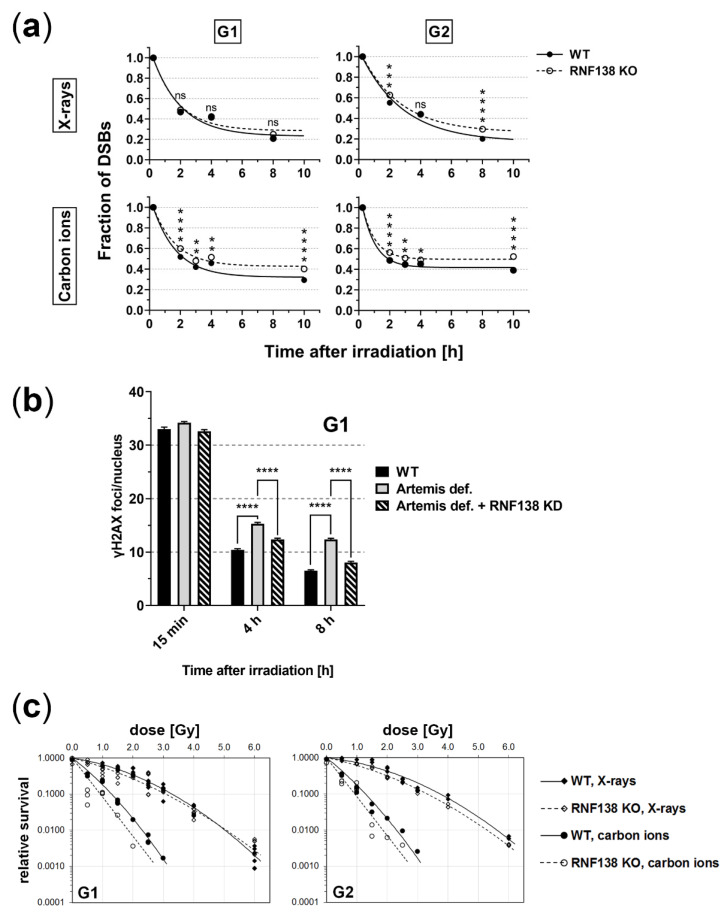
**RNF138 deficiency diminishes DSB repair and survival of heavy-ion-irradiated cells.** (**a**) U2OS cells, RNF138 proficient (wt) or deficient (RNF138 KO), were irradiated with 1 Gy X-rays or carbon ions (11.4 MeV/nucleon) and fixed at different time points after irradiation. Samples were immunofluorescence stained for γH2AX (DSB marker) and γH2AX foci were quantified per nucleus. The γH2AX-foci number of non-irradiated cells was subtracted and the data normalized to the γH2AX-foci number 15 min after irradiation, which was considered the number of DSBs induced by irradiation. The absolute number of γH2AX foci 15 min after irradiation was similar between the tested genotypes. DNA was visualized with DAPI and EdU; these signals were used for cell-cycle specific analyses (see Section 2.6.). *n* = 3 (X-rays); *n* = 1, 2 replicates (carbon ions); error: SEM (small, covered by the markers). A two-tailed *t*-test was performed on the normalized data, *p* values: ns, not significant; * *p* < 0.05; ** *p* < 0.01; *** *p* < 0.001; **** *p* < 0.0001 (**b**) Artemis proficient cells (82-6 hTERT) and deficient cells (CJ179hTERT) as well as Artemis deficient cells that were depleted for RNF138 by RNAi (CJ179hTERT + RNF138 KD) were irradiated with 2 Gy X-rays and fixed at different time points after irradiation. The samples were immunofluorescence stained for γH2AX and the cell-cycle marker CENP-F, and γH2AX foci/nucleus were quantified in G1 cells (CENP-F negative). *n* = 3; error bars: SEM. Brackets indicate comparisons between samples. Two-tailed *t*-test, *p* values: **** *p* < 0.0001 (**c**) Relative clonogenic survival of HeLa.S-Fucci cells, RNF138 proficient (wt) or deficient (RNF138 KO), that were irradiated with different doses of X-rays or carbon ions (11.4 MeV/nucleon). Immediately after irradiation, the irradiated cell populations were sorted cell-cycle specifically, using the HeLa.S-Fucci characteristic cell-cycle specific fluorescence signal. No data were averaged but all measurements are shown (up to three technical replicates per experiment). n = 2 (X-rays), *n* = 1 (carbon ions).

## Data Availability

The original contributions presented in the study are included in the article/Appendix A. Further inquiries can be directed to the corresponding author.

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
