# Peer review of "The Ubiquitin Ligase RNF138 Cooperates with CtIP to Stimulate Resection of Complex DNA Double-Strand Breaks in Human G1-Phase Cells"

_cells, 2022, doi:10.3390/cells11162561_

Round 1

Reviewer 1 Report

Averbeck and et al report that RNF138 is required for the ubiquination and function of CtIP in DSB resectioning upon complex DSB generation in G1 cells.  The authors compared the differential radiation effect of RNF138 using 30 Gy of X-ray, alpha-particles, carbon and iron ions.  The authors determined resectioning processes through RPA and CtIP signals.  They found that loss of RNF138 resulted in delayed repair kinetics and decreased cell survival.  This is due impart to the mis-post-translational modification of CtIP.

The authors did a great job in the methods and materials section.  Overall the authors wrote an interesting paper; however there are points that the authors should consider to assist in the rigor of the experiments. 

Weaknesses:

1)    The authors need to explain and/or show that the number of and the types of complex DSBs generated from 30 Gy of X-ray, alpha-particles, carbon, and iron ions are the same or different from each other.  In particular, is there a difference in complex DSBs generated from alpha vs carbon vs iron?

2)    In Figure 3, the authors argue that RNF8 is not required for resectioning of complex DSBs in G1 cells when exposed to alpha-particles, but that RNF138 is required when cells are exposed to carbon and iron ions.  It is difficult to agree with the authors since (1) the authors did not utilize at least one common cell line in fig a and b, (2) the authors used different radiation when comparing RNF8 vs RNF138.  The authors should show that RNF138 is required for resectiong upon alpha-particle exposure and that RNF8 is not required in carbon and/or iron radiation.

3)    For Figure 3, I would like to see representative images of RNF8 and RNF138 KD/KO conditions that were used to generate the graphs.

4)    Why does the “normal” fibroblasts not have as drastic effect as the cancer cell lines upon RNF138 loss?  Is this due to the different radiation source or is it due to the cell line?  Again, keeping the radiation and/or cell lines consist would alleviate this confusion.  Use of a secondary (preferably a tertiary) cell line would confirm observations and is ideal.

5)    In figure 5, the authors used 30 Gy of X-ray as a condition.  Can the authors elaborate on this especially since in Fig 2 there shouldn’t have been a differential effect of RNF138 presence in ubiquination?

6)    For which DSB repair pathway does RNF138 affect when it comes to repairing complex DSBs in G1 cells?  Is RNF138 required for proper c- or a- NHEJ or other DNA repair pathways?  Does repair pathway activation differ among the different heavy ion radiations (alpha-, carbon, or iron)?

Reviewer 2 Report

Authors previously reported that resection of DNA double-strand breaks (DSB) against clustered DSB occurs not only in G2-phase but also in G1-phase in response to heavy ions or alpha-particles. This paper further investigated the mechanisms underlying the molecular response on resection-dependent DNA repair pathways including ubiquitin ligase RNF138 and CtIP, Overall, the paper is well organized. This reviewer’s comments are described below.

1.     For protein-pull down experiments with TUBE2 in Figure 2, ubiquitin should be detected by using ubiquitin antibody instead of TUBE2-biotin,

2.     In Figure 2a, amounts of Ku80 expression (loading control) are not equal among samples. Authors should explain the reason.

3.     Describe the methods of quantitative analysis for RPA or CtIP positive cells. It is unclear how to calculate the fraction of positive cells.

4.     To confirm whether RNF8 is necessary on removal of Ku80 in DSB resection of G1-phase cells, authors should examine immunofluorescence with Ku80 and 53BP1 antibodies when RNF8 is depleted

5.     Geminin and gamma-H2ax are mouse antibodies. Authods should explain how to distinguish between Geminin and gamma-H2ax staining in positive cells.

Round 2

Reviewer 1 Report

The authors did a great job clarifying the source of ionizing radiation.  I still would have liked to see additional experiments showing how the loss of RNF138 may impact on alternative NHEJ or other DNA repair pathways in the G1 cells.  For example, using the U2OS-EJ2 system developed by Dr. Jermey Stark or have seen that the RNF138 KO G1 cells were sensitive to Parp-1 inh.

Reviewer 2 Report

The authors have addressed my comments.

Author Response

Dear Reviewer #2:

We are glad that we were able to address all open issues and questions in an appropriate way.

Yours sincerely,

Nicole Averbeck

Round 3

Reviewer 1 Report

Thank you, Authors for the clarifications.

Author Response

Dear Reviewer #1:

You are very welcome. We are glad we could address your request to your satisfaction.

Yours sincerely,

Nicole Averbeck